# Analysis of Genetic Factors Defining Head Blight Resistance in an Old Hungarian Wheat Variety-Based Mapping Population

Emese Varga-László [1,2], Katalin Puskás [2], Balázs Varga [2,*] [ID], Zsuzsanna Farkas [2,3], Ottó Veisz [2] and Gyula Vida [2] [ID]

1   Monsanto Hungária Ltd., 1133 Budapest, Hungary; emese.varga-laszlo@bayer.com
2   Agricultural Institute, Centre for Agricultural Research, 2462 Martonvásár, Hungary;
    puskas.katalin@agrar.mta.hu (K.P.); farkas.zsuzsanna@agrar.mta.hu (Z.F.); veisz.otto@agrar.mta.hu (O.V.);
    vida.gyula@agrar.mta.hu (G.V.)
3   Festetics Doktoral School, Georgikon Faculty, University of Pannonia, 8360 Keszthely, Hungary
*   Correspondence: varga.balazs@agrar.mta.hu; Tel.: +36-22-569500

**Abstract:** One of the most important limiting factors of high-quality wheat production is *Fusarium* head blight infection. The various *Fusarium* species not only may cause severe yield loss but—due to toxin production—the grains also might become unsuitable for animal and human nutrition. In the present research, our aim was to examine the Fusarium resistance of a special mapping population ('BKT9086-95/Mv Magvas') and identify the genetic factors and chromosome regions determining the tolerance to *Fusarium culmorum* and *Fusarium graminearum*. The connection between the genetic background and the *Fusarium* head blight sensitivity was confirmed by the analysis of variance in the case of three markers, among which the co-dominant pattern of the gtac2 and gtac3 amplified fragment length polymorphism (AFLP) markers might indicate a marker development possibility. Consistently expressed quantitative trait loci (QTLs) were identified on the chromosomes 2A, 2B, 2D, 5A, and 7A. Loci linked to resistance were identified on 11 chromosomes. During the investigation of phenological and morphological traits (heading date, plant height, ear compactness) influencing the head blight resistance and the location of the resistance QTLs, the total overlap was found in the case of the region identified on chromosome 2D and partial overlap on chromosomes 2A and 2B. Whereas 5A may be a rare allelic variant of a novel QTL.

**Keywords:** biotic stress tolerance; resistance source; mycotoxin; plant breeding; yield quality

## 1. Introduction

Wheat is one of the major staple foods worldwide; however, its yield quantity, as well as quality, may be seriously damaged by various pests and diseases. Among them, special attention has to be paid to the *Fusarium* species since they may threaten yield parameters in various ways. Particularly significant damages are, for example, weak seedling emergence, root and stem rot, as well as head and seed infections; however, the most harmful disease among them is Fusarium head blight (FHB). As a result of the infection, the crop yield may drastically drop, the bread-making quality and the seed vigor may deteriorate. Between 1990 and 2000, Fusarium epidemics caused approximately 2500 billion USD yield loss in the US only [1]. In many parts of the world, due to climate change, the conditions are becoming more and more favorable for the emergence of an epidemic [2,3]. In addition to the qualitative and quantitative losses, the *Fusarium* species produce mycotoxins as secondary metabolites (deoxynivalenol, nivalenol, zearalenone, T-2 toxin, etc.), which present health risks both for humans and livestock. Therefore, if their amount is above the risk threshold, the crops become

unsuitable for processing [4]. Their presence is particularly detrimental and hazardous, as currently, there is no solution for their subsequent removal from the infected grains. Besides, their molecules are thermostable, and thus they cannot be expected to decompose during processing either [5].

There are several possibilities to mitigate the yield loss caused by Fusarium infections in cereal crops. Some agronomic treatments might be beneficial for reducing the pathogen pressure, and even the morphology of the produced variety, such as plant height and ear compactness, could influence the rate of the contamination [6,7]. Based on previous researches, it can be stated that the negative impacts of the Fusarium head blight infection can be tackled the most effectively by using resistant genotypes [2,3]. Fortunately, large genetic variation for FHB resistance is available in the wheat gene pool [8,9]; however, the best regionally adapted and highly productive cultivars are often susceptible to FHB. Initially, the determination of FHB resistance genes was focusing on East Asian genotypes ('Sumai-3', [10]); however, these are spring wheat genotypes and have special agronomic traits; therefore, it is difficult to involve them in the conventional breeding programs in Central Europe [11,12]. As germplasm screening gradually became a focus of interest, further resistance sources were identified in South America (e.g., 'Frontana', [11,12]), Europe [13,14], and in North America where new elite lines were released based on the already known resistance sources (e.g., Alsen), and local, adapted cultivars were also recorded (e.g., Emerson) [15–17].

The resistance itself has more components, such as resistance to the initial infection (type I), spread from the infected florets (type II), resistance to kernel infection (type III), and tolerance and resistance to toxin accumulation (type IV and V) [18,19]. Furthermore, the genes influencing the Fusarium resistance descend as quantitative properties; therefore, the gene expression is determined through the interaction of many genes and QTLs (quantitative trait loci), and thus the resistance of the descendant generations can be represented on a continuous scale [20,21].

Fusarium head blight is known in Hungary already since the 1920s. Still, the first nationwide epidemics occurred only in the 1970s, which can be explained by the susceptibility of the varieties in addition to the spreading of intensive production technologies and the weather conditions favorable for the infection. The genetic background of the varieties used in large-scale cultivation also might have contributed, with great probability, to the insignificant cases of infections between the 1920s and the 1970s—nationwide epidemics were unknown during this period [22].

The objective of this research was to assess the resistance to Fusarium head blight and to identify the genetic factors and chromosome regions encoding the trait. For this purpose, the analysis of a population established by crossing a line derived from an old Hungarian wheat variety and a modern wheat cultivar bred at the Center for Agricultural Research (CAR) in Martonvásár was carried out.

## 2. Materials and Methods

### 2.1. Plant Material

The mapping population consisting of 250 lines was established by the SSD method (single seed descent [23]). The parents were selected based on their reactions to FHB. Fusarium head blight resistance of several lines originating from the heterogeneous population of 'Bánkúti 1201' was analyzed previously in artificially inoculated field trials in Martonvásár. 'Bánkúti 1201' is an old Hungarian winter wheat variety bred in 1931. Apart from the typical unfavorable traits of old wheat varieties (tall plant height, prone to lodging, susceptibility to leaf diseases), it has outstanding storage protein composition and, consequently, excellent bread-making quality [24]. Owing to its prosperous characteristics, the variety still has relevance in Hungary, primarily, in organic production [25]. Among the tested lines, 'BKT9086-95' proved to be consistently resistant to Fusarium head blight at the same level as the resistant control variety ('Sumai 3'); therefore, this line was selected as the resistant parent of the mapping population.

The winter wheat variety 'Mv Magvas' proved to be susceptible during the routine determination of Fusarium head blight resistance carried out as part of the breeding processes in Martonvásár. Therefore, it was selected as the susceptible parent.

To determine the resistance to the fungal spread within the head (type II resistance, [18]), the FHB resistance tests were initiated at the $F_5$ generation of the mapping population.

## 2.2. Artificial Inoculation

The tests were carried out under controlled greenhouse conditions and under irrigated field conditions. The same artificial inoculation method was applied in both systems. IFA66 *Fusarium graminearum* and IFA104 *F. culmorum* isolates were used for the examinations. Sterilized soil/sand mixture was used for the long-term storage of the isolates. The monoconidial culture was cultivated in SNA (synthetic nutrient-poor agar [26]) medium.

In the case of *F. graminearum*, the necessary amount for the artificial inoculation was obtained by multiplication in mung bean liquid medium [27], and in sterilized wheat-oat seed mixture in the case of the *F. culmorum* [28]. Surfactant and mycelia were removed and discarded by a vacuum pump from the medium solution of *Fusarium graminearum*, and after filtration, the concentration of conidia was determined. In the case of *Fusarium culmorum*, distilled water was used to wash out the macroconidia from the wheat-oat mixture, and the concentration of the suspension was determined.

The same method was used for the artificial inoculation under greenhouse and field conditions. In the greenhouse, 4 ears of each line were injected with *Fusarium culmorum*, and in the field, 5 heads were inoculated by *Fusarium graminearum* and *Fusarium culmorum* isolates. The conidia concentration was set to $5 \times 10^5$/mL in the case of both greenhouse and field infections. The main ear of each plant was artificially inoculated with the *Fusarium* isolates (BBCH 61). At the upper 1/3 of the flowering head, 5 μL conidial suspension was injected into the two basal florets of a spikelet [29,30] using a repeating pipette, with wounding. The inoculated heads were marked with a self-adhesive label affixed to head supporting the stem segment. Color-coding was applied to distinguish the pathogens.

The evaluation was carried out according to the same method, both under greenhouse and field conditions. Simultaneously with the infection, the total number of spikelets was recorded. Progression of the symptoms was assessed on the 7th, 14th, and 21st days after inoculation (DAI), and the number of infected florets was determined.

Experiments were conducted over 3 years (2007–2009) under controlled conditions in the experimental greenhouse of the Agricultural Institute, Centre for Agricultural Research. Plants were germinated, and after vernalization (42 days at 4 °C), they were planted into 2-liter pots (12 cm diameter), one by one. The length of the illumination was set to 16 h, and the temperature of the chambers was regulated by the Spring-Summer climatic program, which was designed to simulate the typical climatic conditions of Hungary [31]. In order to create favorable conditions for the *Fusarium* infection, from the heading to the end of the flowering stage (BBCH 51–69, [32]), mist irrigation was used to elevate the relative humidity to 90%. The tests were conducted in four repetitions in each experiment.

Field studies were carried out in the Fusarium head blight nursery of the Centre for Agricultural Research (CAR), in three growing seasons: 2005/2006, 2008/2009, and 2010/2011 (47°18'47''N, 18°46'24''E). Two 2-meter long rows were sown in the first decade of October with a HEGE-80 drill (Hege Ltd., Ladenburg, Germany), in each year. The distance between the rows was 20 cm. The soil type was sandy loam soil with a depth of 100 cm humic layer. The applied agronomic treatments were based on the local conventional practice.

Before sowing, 150 kg·ha$^{-1}$ complex fertilizer was used (N:P:K = 2:1:1), and in the spring, an additional 60 kg·ha$^{-1}$ of N was applied as top-dressing. As for plant protection, herbicide 2,4-D and insecticide esfenvalerate were sprayed out two times in March and April. From the beginning of the heading, the plots were mist irrigated to ensure optimal humidity for the *Fusarium* infection. In this phase, flowering date, plant height, and length of the heads were also recorded.

The meteorological conditions of the experimental years are displayed in Figure S1, showing long-term average data (1981–2010).

## 2.3. Statistical Analyses

Infection severity 21 days after inoculation (21 DAI) was determined as a percentage of the total spikelet number. The area under the disease progress curve (AUDPC) [33] was also calculated; from these data, conclusions could be drawn about the progress of the infection over time [19].

The statistical assessment of the data was carried out by the R programming packages [34]. In accordance with the requirements of the parameter tests, the distribution of the samples was tested by the Shapiro–Wilk method, and the homogeneity of variance was checked by the Levene test. The analysis of variance (ANOVA) and correlation calculation was applied for testing the relationships between 21 DAI, AUDPC, and plant height, inoculation time (beginning of flowering), and ear compactness [35,36]. In the case of unequal sample size, unbalanced ANOVA was performed (in 2009, under field conditions, only the *Fusarium graminearum* inoculation was evaluated). As there is no species-specific FHB resistance, the fungal isolates were not considered as factors [19].

## 2.4. Molecular Methods

### 2.4.1. DNA Extraction

DNA was extracted from the plant samples of 250 lines and the parents using Qiagen DNeasy Plant Mini Kit (Qiagen, Hilden, Germany) according to the instructions of the manufacturer (from generation $F_6$, parallel with the first greenhouse testing form the 5th non-infected repetition). The DNA concentration of the samples was determined by NanoDrop 1000 (Thermo Fisher Scientific, Waltham, MA, USA). The templates were stored at $-20\,°C$ until further processing.

### 2.4.2. DNA-Based Markers (SSR, AFLP, SNP)

As a first step, the differences between the parents were analyzed with the selected SSR and AFLP primers [37–39]. The selected SSR markers covered the wheat genome in 20–40 cM intervals, and those that revealed polymorphism were tested at the whole population level. The separation of the reaction products and the detection of the samples were performed for the SSR markers using the Li-Cor 4200 (Li-Cor Biosciences, Lincoln, NE, USA) instrument on 6% polyacrylamide gel. The AFLP fragments were separated on 7% polyacrylamide gel, and the patterns were analyzed using the Typhoon Trio™ (GE Healthcare, Chicago, IL, USA) system (at 520 nm, 570 nm, and 670 nm wavelength).

The genotypes that had full data series in the greenhouse from each year and each repetition were also analyzed by Illumina Infinium (TraitGenetics GmbH, Gatersleben, Germany) 20k wheat chip. From the 17,262 markers, 5528 were polymorphic, after excluding those which were deviated from the 0.4–0.6 segregation ratio, 4263 SNP markers were used for the association analysis.

### 2.4.3. QTL Identification

Marker-trait association was performed on a genetic database based on AFLP and SSR markers. The possibility of a relationship between the markers and Fusarium head blight was analyzed by the program package GAPIT—Genome Association and Prediction Integrated Tool—based on the position of the SNP markers in the pseudo-reference genome (iwgsc refseq v2.0) [40,41].

Since the lines originated from an experimental cross, the 'K' model was chosen, which handles the variance-covariance matrix as random, treating each line as a separated group [42]. Manhattan plots of $-\log10$ ($p$) values for each SNP vs. chromosomal position (iwgsc refseq v2.0) were generated as the GAPIT results. The SNPs with $-\log10$ ($p$) > 2.5 or $p < 7.9 \times 10^{-3}$ were considered to be significant [43].

The individual lines were characterized by the average values of the greenhouse and field data series, as well as the BLUP (best linear unbiased prediction) values applied to the total test [44]. The BLUP values were calculated with the program package lme4 [45].

In the different experimental locations and years, BLAST (basic local alignment search tool) characterization was carried out based on the known sequence of the Affymetrix markers with significant effect.

## 3. Results

### 3.1. Review of Type II Resistance of the Offspring Lines Originating from the 'BKT9086-95/Mv Magvas' Experimental Cross under Greenhouse Conditions

Regarding the percentage of infection 21 DAI, the values of the wheat lines making up the population covered the entire scale of infection. The highest average infection was observed in 2009 (65.2%); however, a similar average value (52.95%) was inspected in 2007 too. In contrast, the average infection was as low as 24.17% in 2008 (Table 1).

**Table 1.** Annual average infection of lines and parental genotypes under greenhouse conditions ($n = 175$, Martonvásár, 2007–2009).

| | Head Infection (%) | | | | |
|---|---|---|---|---|---|
| | **Parents** | | **Lines ($n = 173$)** | | |
| | 'BKT9086-95' | 'Mv Magvas' | Average | Range | Standard Deviation |
| 2007 | 32.25 | 95.24 | 52.95 | 4.09–95.91 | 24.75 |
| 2008 | 12.00 | 94.43 | 24.17 | 5.15–96.45 | 17.99 |
| 2009 | 38.06 | 97.52 | 65.20 | 3.85–100.00 | 25.44 |

It was found that each year, the average infection values of the lines were almost two times higher than the infection values of the resistant parent ('BKT9086-95'). Moreover, the pathogen could reach almost every spikelet in the head of the susceptible parent and trigger symptoms therein ('Mv Magvas').

The two-factor analysis of variance of the infection values at $p < 0.001$ level also showed a significant difference between the genotypes and infection severity with respect to both the infection percentage and the AUDPC values (Table 2). At the same time, it can be stated that the experimental year had an impact on the rate of *Fusarium* infection 21 DAI with statistical proof.

**Table 2.** Result of the analysis of variance based on the infection values 21 DAI and the AUDPC data (21 DAI: infection severity % on the 21st day after inoculation; AUDPC: area under the disease progress curve), under greenhouse conditions in 2007 and 2009 ($n = 173 + 2$ parents).

| | **21 DAI** | | | |
|---|---|---|---|---|
| | **SST** | **Df** | **F-value** | **Pr > F** |
| Genotype | 448,802 | 174 | 4.001 | $1.45 \times 10^{-10}$ *** |
| Year | 5751 | 1 | 2.461 | 0.035 * |
| Genotype × Year | 123,863 | 174 | 0.552 | 1.000 |
| Residual value | 1,026,146 | 796 | | |
| | **AUDPC** | | | |
| | **SST** | **Df** | **F-value** | **Pr > F** |
| Genotype | 51,294,274 | 174 | 2.361 | $1.2 \times 10^{-15}$ *** |
| Year | 50,592 | 1 | 0.405 | 0.525 |
| Genotype × Year | 16,902,477 | 174 | 0.778 | 0.979 |
| Residual value | 99,403,034 | 796 | | |

SST: the total sum of squares, Df: the degree of freedom, Pr > F: probability. Significance levels: '***' 0.001; '*' 0.05.

Our greenhouse results supported the presence of the genetically determined Fusarium head blight resistance in the tested population. A QTL with appropriate effect, which is also useful for practical breeding, expresses under different environmental conditions as well. Therefore, detailed field tests were conducted on the population to determine this aspect.

*3.2. Review of Type II Resistance of the Offspring Lines 'BKT9086-95/Mv Magvas' under Field Conditions*

In the case of infection severity (%), the values of the wheat lines making the population varied on a wide scale under field conditions too (Table 3), covering the total scale of infection each year. The strongest average infection was inspected in 2011, in both *F. culmorum* (53.16%) and *F. graminearum* isolates (65.95%), while the lowest average infection was observed in 2006 (25.27%).

**Table 3.** Average field infection values of the lines and parental genotypes (*n* = 223, Martonvásár, 2006, 2009, 2011).

| | | \multicolumn{2}{c|}{Parents} | | \multicolumn{3}{c}{Lines (*n* = 221)} | | |
| | **Isolate** | **'BKT 9086-95'** | **'Mv Magvas'** | **Average** | **Range** | **Dev.** |
|---|---|---|---|---|---|---|
| **2006** | Fc | 6.67 | 100.0 | 30.90 | 4.35–76.90 | 18.14 |
| | Fg | 8.51 | 92.86 | 25.27 | 4.35–92.56 | 14.14 |
| **2009** | Fg | 26.19 | 100.0 | 47.10 | 7.01–91.67 | 17.49 |
| **2011** | Fc | 25.53 | 72.75 | 53.16 | 8.95–100.00 | 18.83 |
| | Fg | 23.99 | 85.99 | 65.95 | 9.93–100.00 | 20.62 |
| **Ave.** | Fc | 16.10 | 86.38 | 42.03 | 4.37–100.00 | 18.22 |
| | Fg | 19.56 | 92.95 | 46.10 | 9.93–100.00 | 20.74 |
| | Main average | 18.18 | 89.67 | 44.48 | 11.19–90.83 | 12.99 |
| | PH | 125.0 | 85.00 | 115.6 | 85.00–142.50 | 11.74 |
| | Heading | 19.0 | 21.00 | 21.60 | 14–30 | 2.51 |
| | Ear comp. | 1.67 | 2.50 | 2.10 | 1.53–2.78 | 0.28 |

Fc: *F. culmorum*, Fg: *F. graminearum*, PH: plant height (cm), Heading: number of days from 1 May, Ear comp.: ear compactness (number of spikelets/cm), Dev.: deviation, Ave.: average (note: in 2009, only Fg inoculation was conducted).

It was found that the positions of the parental genotypes on the scale corresponded with their known resistance each year. Genotypes less infected than the resistant parent were identified each year. The susceptible 'Mv Magvas' was located at the endpoint of the scale, except for the year 2011, which was characterized by the highest average infection. The most important result was that—similarly to the greenhouse results—such genotypes were identified in which the *Fusarium* species could not spread to the adjacent spikelets from the point of injection.

The most accurate understanding of Fusarium head blight resistance could be obtained by analyzing the multi-year data series. Therefore, the effects of genotype, year, time of infection, plant height, and genotype × year interaction on the infection severity values were also tested with respect to the whole experiment. As different numbers of repetitions were used, throughout the years, we applied an unbalanced ANOVA test for the statistical evaluation of the 3-year data series (Table 4). The correlation between the phenotypic traits and resistance and the distribution can be found in Figure S2.

Supported by statistical evidence, it was found that the genotype, year, and date of inoculation had an effect on both the infection values 21 DAI and on the area under the disease progress curve. Regarding the whole experiment, plant height and ear compactness had a significant impact on infection severity; however, the AUDPC values were not affected by them. This difference showed that despite the close interrelationship between the two indices, they might provide supplementary information for characterizing the level of resistance.

**Table 4.** The results of the unbalanced ANOVA test, on the basis of the 21 DAI and AUDPC values (21 DAI: infection severity % on the 21st day after inoculation; AUDPC: area under the disease progress curve), under field conditions, based on the years 2006, 2009, and 2011 ($n = 221$).

| 21 DAI | | | | |
|---|---|---|---|---|
| | **SST** | **Df** | **F-value** | **Pr > F** |
| Genotype | 609,534 | 220 | 2.9057 | $2.2 \times 10^{-16}$ *** |
| Year | 55,986 | 2 | 5.4406 | $6.829 \times 10^{-7}$ *** |
| Date of infection | 37,636 | 12 | 3.3491 | 0.019742 * |
| Plant height | 5095 | 16 | 3.7365 | $7.247 \times 10^{-5}$ *** |
| Ear compactness | 12,525 | 25 | 3.3436 | 0.009695 ** |
| Genotype × Year | 190,936 | 192 | 1.0619 | 0.272554 |
| Residual value | 685,810 | 2868 | | |
| **AUDPC** | | | | |
| | **SST** | **Df** | **F-value** | **Pr > F** |
| Genotype | 39,177,695 | 220 | 2.8577 | $2 \times 10^{-16}$ *** |
| Year | 1,875,384 | 2 | 1.9151 | 0.01534 * |
| Date of infection | 1,610,250 | 12 | 2.1925 | 0.0994 * |
| Plant height | 28,482 | 16 | 0.4654 | 0.49518 |
| Ear compactness | 385,892 | 25 | 1.5763 | 0.17782 |
| Genotype×Year | 12,957,508 | 192 | 1.1027 | 0.16585 |
| Residual value | 75,532,613 | 2868 | | |

SST: the total sum of squares, Df: the degree of freedom, Pr > F: probability. Significance levels: '***' 0.001; '**' 0.01; '*' 0.05.

Moreover, considering the related *p* and F values, it can be stated that the year had a smaller influence on the AUDPC value than on the infection severity on the 21st day. Based on the result of ANOVA, no statistical evidence could be found to prove the effect of genotype × year interaction and in the case of the other analyzed traits. This shows that while the year also had an impact on the degree of Fusarium head blight resistance, the order of the genotypes among the specific years did not change in a statistical sense. Consequently, the genotypes of the tested population might indeed carry a genetically defined FHB resistance that could be identified by molecular methods since the variability of the population is adequate for such analysis.

### 3.3. Molecular Tests in the 'BKT9086-95/Mv Magvas' Offspring Population

For the testing of the whole population, those markers were primarily selected, which had known linkage to FHB resistance based on literature data and those that produced signals at least at a medium intensity. As a result, the detailed analysis of the lines was performed with 33 SSR primer pairs. The whole population was tested with 32 AFLP primer combinations as well. Accordingly, 286 polymorphic markers were identified. As the genetic map, which could be created by using the available 319 polymorphic markers, would not cover all chromosomes of the whole wheat genome, ANOVA test was performed to analyse the association between Fusarium head blight resistance and the markers. Using the AFLP and SSR markers, parallel tests were conducted under greenhouse and field conditions to reveal the relationships between the other resistance-related phenotypic traits.

As the genetic map-based QTL analysis provided statistically more reliable results than the analysis of the marker-trait associations, only those markers were considered as linked that had a significant effect within the specific experimental systems in all experimental years and also in comparison with the average. Altogether 30 markers were identified that fulfilled the above criteria (29 AFLP, 1 SSR). As during the analysis of the phenotypic data, a significant correlation was found between plant height, ear compactness, flowering time, and the expression of FHB resistance, only those markers were considered as actually resistance-related, which were not influenced by the above-mentioned properties.

Altogether six markers were identified, which had a significant effect on the 21 DAI in both experimental systems. On the basis of the results, we can conclude that agat17, gtac2, and gtac3 markers might be related to the region encoding exclusively the resistance to *Fusarium* spread in the wheat head.

The ANOVA test was also performed in the case of the AUDPC values, as a result of which 25 markers were identified either under greenhouse or field conditions, which were related to the AUDPC values of the lines, supported by statistical evidence. The partial overlap could be detected among the markers that were in significant connection with the 21 DAI and the AUDPC. Among them, 14 markers were found to be connected to both values. However, the markers exclusively connected to the AUDPC values might also be related to other phenotypic traits without exception.

On the other hand, supported by statistical evidence, the AFLP markers agat17, gtac2, gtac3 (the latter two showing codominant nature) also had an effect on the AUDPC. Based on this, it can be stated that the 'BKT9086-95/Mv Magvas' offspring lines might carry genetically determined FHB resistance with high probability.

### 3.4. Identification of the Genetic Background Related To Infection Severity

For the characterization of the lines, we used the average values of the greenhouse and field data series. Based on the K model of the GAPIT program package, the presence of significant QTLs was confirmed under greenhouse conditions in relation to the infection values 21 DAI (Figures 1 and S3, and the significant SNP markers are displayed on Table S1). The connection between 32 markers was identified with lower infection severity, all together on five chromosomes. The lowest *p*-values were calculated for the chromosome 7A: $p = 0.00026$ (Ra_c8394_1381) and $p = 0.00442$ (BobWhite_c30461_131) in the nucleotide positions 645,092,185 and 647,933,749, respectively, on the physical map of the wheat. A total of 14 markers in a significant relationship with the infection values 21 DAI were identified on the chromosome 7A under greenhouse conditions.

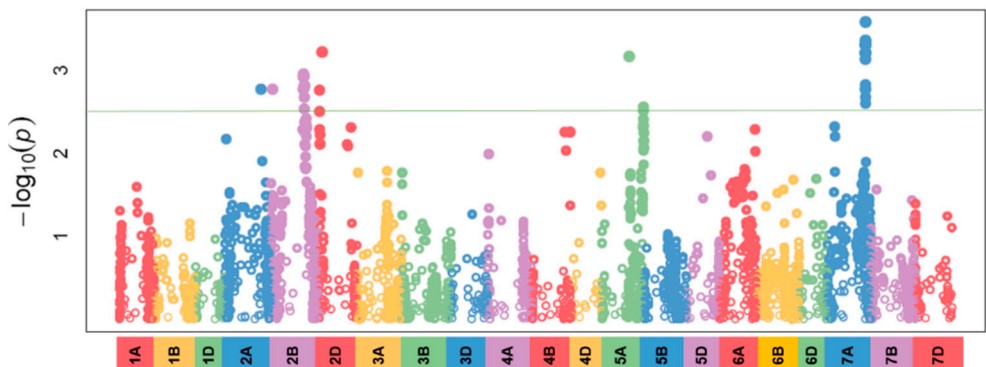

**Figure 1.** Manhattan plot, average infection values on the 21st day after inoculation under greenhouse conditions, according to the chromosomal localization of the markers (level of significance: -$\log_{10}(p)$ 2,5; Martonvásár, 2007, 2009).

On the short arm of the chromosome 2B (Tdurum_contig29563_257, pos: 28339816), one, while on the long arm, 10 markers revealing close correlation with the resistance were identified (the region between markers AX-94393508 and BobWhite_c16130_362). In the case of chromosomes 2D and 5A, three additional, significantly linked markers were found in the nucleotide regions 15,967,374–62,023,977 and 466,617,397–702,461,388, respectively. A single SNP marker with significant effect was identified on the chromosome 2A (RAC875_c29716_871, pos: 607997395).

Based on the estimation of the specific alleles' effects, the QTL region on chromosome 7A originated from the susceptible parent ('Mv Magvas'), while these regions on chromosomes 2A, 2B, 2D, and 5A were connected to the resistant parent ('BKT9086-95').

Genetic regions with significant effects could be identified in a smaller number by projecting the infection percentage to the whole field trial. The linkage could be identified in the case of six markers in total, on the chromosomes 2A, 2D, and 7A (Figures 2 and S3). On chromosome 2A, three significantly linked markers were identified, among which RAC875_c29716_871 indicated the presence of genetically determined resistance even under controlled conditions. One region with significant effect was identified on chromosome 2D and two on chromosome 7A. In contrast to the greenhouse tests, under field conditions, no linkage could be confirmed for genetically determined resistance on chromosomes 2B and 5A.

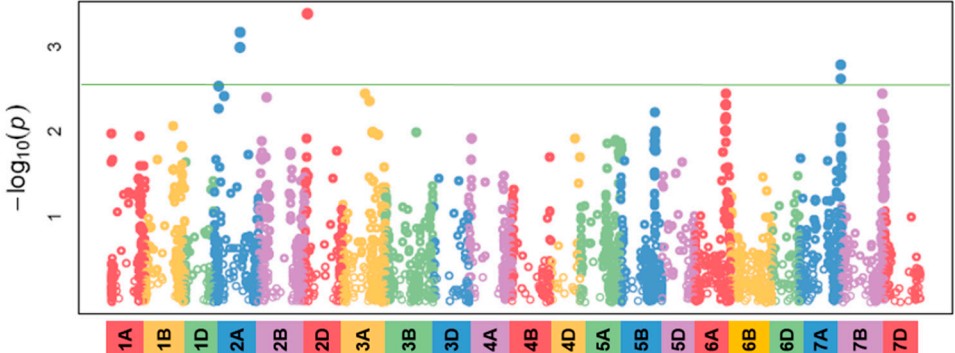

**Figure 2.** Manhattan plot, average infection values on the 21st day after inoculation under field conditions, according to the chromosomal localization of the markers (level of significance: -$\log_{10}(p)$ 2,5; Martonvásár, 2009, 2011).

During the analysis of quantitative properties, the characterization of the lines using average values may hide the presence of low-effect QTLs. Therefore, the analysis was also performed with the BLUP values of the lines projected to the whole trial (Figures 3 and S3). As a result, 37 linked markers were identified on five chromosomes. The largest group of markers was identified on chromosome 5A: 16 markers were found in the region between markers wsnp_Ku_c38543_47157828 and wsnp_Ex_c2171_4074003, while the marker AX-94978476 (pos: 466617397) was identified in a different region. A group of 11 markers was identified on chromosome 7A, in the region between markers AX-94976788 and Kukri_rep_c98227_390. The presence of a group consisting of five markers was detected on chromosome 2B in the position between markers AX-94393508 and AX-94507617. On chromosome 2D, a group of four markers was identified, while on chromosome 2A, we could detect linkage only in the case of one single marker (RAC875_c29716_871, pos: 607997395).

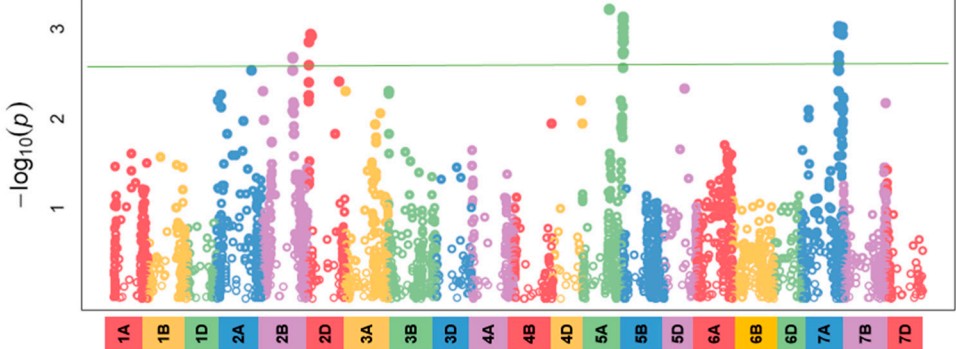

**Figure 3.** Manhattan plot, best linear unbiased prediction (BLUP) values of infection on the 21st day after inoculation, across years and study sites, according to the chromosomal localization of the markers (significance level: -$\log_{10}(p)$ 2,5; Martonvásár, 2006, 2007, 2009, 2011).

A special characteristic of the loci determining the quantitative traits with the small and medium effect is that their detection under different environmental conditions is not always possible. Therefore, the correlations between the FHB resistance and the genotypes were analyzed on a yearly basis as well. Regarding the whole experiment, the presence of genetic regions with an effect on the spread of *Fusarium* in the head was confirmed on 12 chromosomes.

### 3.5. Identification of the Genetic Background In Correlation With the Size of the Area Under the Disease Progress Curve

Under greenhouse conditions, altogether 37 markers were identified in six chromosomal regions, which showed a significant correlation with the average size of the area under the disease progress curve (Figures 4 and S3, and the significant SNP markers are displayed in Table S2). Correlations could be detected on chromosomes 2A, 2B, and 5D in the case of one marker each. Compared to the infection values 21 DAI, a difference was represented by the significant QTL presence in the 349,866,506 nucleotide position of chromosome 5D, which, however, was also of 'BKT9086-95' origin just like the QTLs of chromosomes 2A, 2B, and 2D. A group of seven markers was identified on chromosome 2D in the position between markers AX-94908406 and AX-95124335. On chromosome 5A, 16 significant marker-trait associations were identified. The closest correlation was observed in the case of marker AX-94978476 ($p = 0.0004$), which is located in the position 466,617,397 of the nucleotide of the pseudo-chromosome. On the chromosome, 15 additional markers were categorized into one group in a significant correlation with the size of the area under the disease progress curve, in a different region regarding the chromosomal localization. In the case of chromosome 7A, 11 markers in a significant correlation with the analyzed trait were identified. Based on the localization of the markers, it is assumed that two areas are affected. The region identified on chromosome 7A originated from the genetic background of the susceptible parent, also in the case of the AUDPC values.

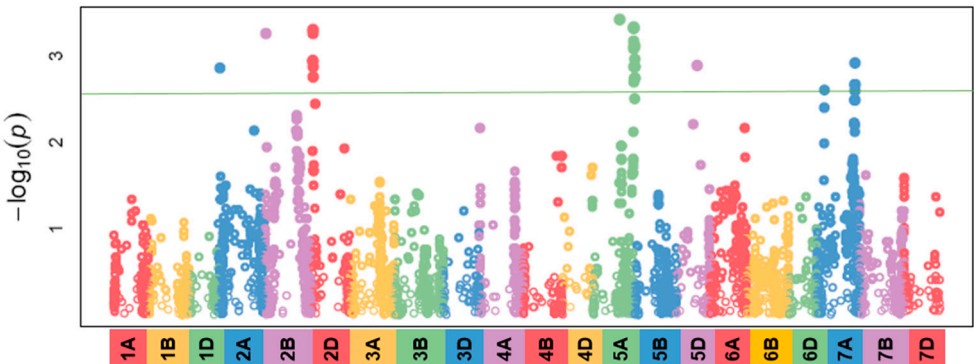

**Figure 4.** Manhattan plot, average area under the disease progress curve (AUDPC) values under greenhouse conditions, according to the chromosomal localization of the markers (level of significance: -$\log_{10}(p)$ 2,5; Martonvásár, 2007, 2009).

Under field conditions, just like in the case of infection 21 DAI, the analysis of the average AUDPC values (Figures 5 and S3) revealed a significant correlation for fewer markers only. A total of six markers could be detected on five chromosomes with a clear correlation between the analyzed traits.

However, the chromosomal localization of the Tdurum_contig915119_224 and AX-94551829 markers identified on the chromosome 2A differed from the QTL position identified under greenhouse conditions. A significant QTL effect could be observed on the chromosomes 2D and 7A under field conditions as well. At the same time, no QTL could be confirmed on the chromosomes 2B and 5A under field conditions contrary to the greenhouse observations. However, the QTL region identified on chromosomes 3A and 6A could be detected under field conditions only.

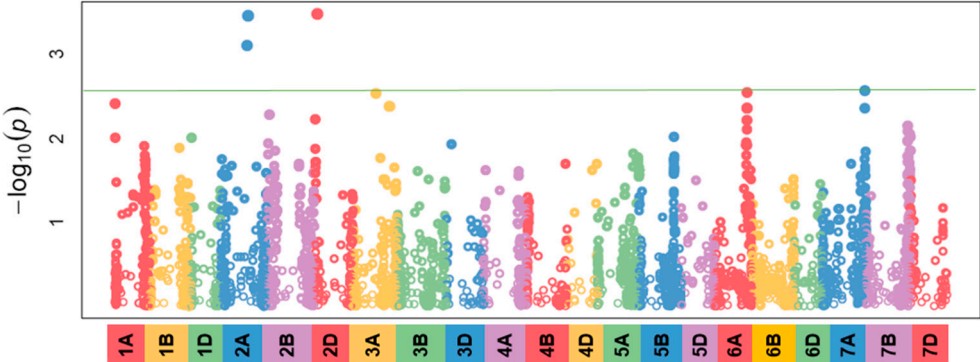

**Figure 5.** Manhattan plot, average AUDPC values under field conditions, according to the chromosomal localization of the markers (level of significance: -log$_{10}$(*p*) 2,5; Martonvásár 2006, 2009, 2011).

In the case of the area under the disease progress curve as well, the chromosomes were analyzed for the presence of QTL with respect to the BLUP values, with reference to the whole experimental system (Figures 6 and S3).

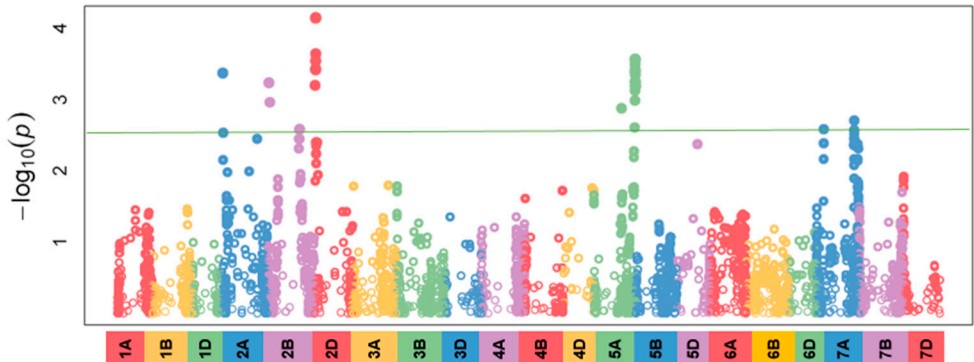

**Figure 6.** Manhattan plot, AUDPC BLUP values, across all experimental years and sites, according to the chromosomal localization of the markers (significance level: -log$_{10}$(*p*) 2,5; Martonvásár 2006, 2007, 2009, 2011).

A total of 35 linked markers were identified on the chromosomes 2A, 2B, 2D, 5A, and 7A. The presence of locus defining quantitative traits was identified in the greatest number on chromosomes 5A and 7A. The region coding the resistance originated from the genetic background of the parent 'BKT9086-95' in the case of chromosome 5A.

As with the infection values, the analysis of the marker-trait associations was performed in the case of the area under the disease progress curve as well, broken down by year. Significant marker-trait associations could be detected in the case of 11 chromosomes at least in one experimental year; however, in one-half of the cases, this correlation could not be detected consistently throughout all experimental years and systems.

### 3.6. Comparison of the Genetic Regions Related To the Infection Percentage and the AUDPC Values

In the case of both infection 21 DAI and the AUDPC values, such genetic regions were identified on five chromosomes, which proved to have a significant effect in several experimental years and systems (2A, 2B, 2D, 5A, 7A). On the basis of the ANOVA results, the environmental factors had a smaller impact on the AUDPC values of the lines, and, therefore, they can be considered as a more reliable indicator. All the above findings are also supported by the results of the marker-trait association analysis, as the size of the genetic region linked to the AUDPC values was restricted to a narrower region in each case. The only exception was the region found on chromosome 5A, where a slight

deviation was detected in the number of markers linked to two resistance metrics for the benefit of AUDPC. Among the identified QTLs, the region detected on chromosome 7A originated from the genetic background of the susceptible parent.

*3.7. Linkage of Resistance to Head Blight and Other Phenotype Properties*

The QTLs linked to Fusarium head blight resistance were frequently located in a position related to other phenotypic traits. Plant height, heading time, as well as ear compactness also had a significant effect on the disease symptoms recorded 21 DAI based on the ANOVA tests. At the same time, the AUDPC values were influenced only by the flowering time. The locus related to plant height was identified on the chromosomes 2A, 2D, 4A, 4B, 4D, and 6A, among which 2A and 2D showed partial overlap with the QTLs defining FHB resistance. In the case of the markers showing the overlap, the favorable allele was carried by the resistant parent. As a result of the correlation test, it was found that the taller plants showed less severe symptoms, and the disease progressed at a slower rate. Regarding ear compactness, statistically proved correlation could be found on chromosomes 2A, 2B, and 2D, and their positions were largely identical with those loci determining plant height.

The genetic region linked to the flowering time was also identified on chromosomes 2A, 2B, and 2D; however, they were located on a different area than the regions that determine plant height and ear compactness. The resistant parent was heading and flowering in an average of 2 days earlier than the population. Based on the ANOVA results and the correlation analysis, the early development represented an advantage in the examined population with respect to head blight resistance.

## 4. Discussion

The success of QTL detection is highly influenced by the accuracy of the phenotyping methods since the trait itself is under the influence of various environmental factors and morphological traits [46]. Single floret inoculation is a widely used method for testing resistance against fungal spread within the head (type II resistance) [8].

Based on our previous results and our three-year investigation [47], the field resistance of 'BKT9086-95' did not differ significantly from the known resistance source 'Sumai 3'. However, according to our results, the resistant parent could be characterized as intermediate in terms of FHB resistance under greenhouse conditions. Moreover, under greenhouse conditions, significantly higher infection severity values were observed compared to the field trials based on the average of the lines and parents as well. The reason behind our observation is probably that under field conditions, the phenotypic traits could interact with the various environmental factors, and thus escaping mechanisms were observed instead of genetically coded resistance [48,49].

QTLs detected on chromosomes 2A, 2B, 2D, and 5A were associated with both the FHB infection severity on the 21st day and the area under the disease progress curve. For comparing our findings with the results of other studies on FHB resistance, the McIntosh Catalog of gene symbols for wheat served as a benchmark [50].

On the chromosome arm 2AL, a QTL with a small effect was detected [51] in the 'Sumai-3/Stoa' mapping population. The resistance-conferring region originated from the background of the susceptible parent 'Stoa'. The presence of the detected QTL was further proved by Paillard et al. [52]; however, the QTL was not stable across years and locations in the 'Arina/Forno' population. Those findings are in line with our results; however, according to the AX-94551829 SNP, the allele type identified at the given position belongs to the 'Arina' type with a frequency of 96.7%, whereas the one identified by the present research belongs to the remaining 3.3%.

There are two QTLs cataloged on chromosome 2B so far. The *Qfhs.crc-2BL* originated from 'Strongfield' durum wheat variety ($R^2$ = 26%), conferring type II resistance [53]. Another QTL is located on the chromosome arm 2BS. It was identified in the 'Renan/Recital' population ($R^2$ = 12%) [54]. Those results are in agreement with our findings; however, the accessible sequence information does not allow us a more detailed comparison.

Due to a small-effect QTL located on chromosome 2D, which originates from the susceptible parental genetic background ([55], *QFhs.pur-2D*), the QTL identified by the present research might be associated with passive resistance mechanisms. The dwarfing gene *Rht8* (present in the susceptible parent 'Mv Magvas') is found on chromosome 2D. Therefore, in this case, it can be presumed that its presence had an unfavorable impact on the head blight resistance. As the pathogen was directly injected into the floret, the effect of the plant height influencing the microclimate was less prominent. Although correlation could be detected between the plant height and the FHB resistance, at the same time, it is assumed that the linkage is based not only on the effect of the phenotype.

Based on previous researches [8], there is a QTL present on chromosome 7A, carried by Chinese germplasms and the 'Ritmo' variety. The presence of *Qfhs.fcu-7AL* QTL was also confirmed in durum variety *Triticum turgidum* ssp. *dicoccoides* substitution lines [56]. Taking into account the origin of 'Mv Magvas' variety and the QTL location described in the present manuscript, it is suspected that a novel QTL was identified even in the genetic background of the susceptible parent.

Chromosome 5A harbors one of the most often studied resistance QTLs, and its presence was identified in Asian, South and East American, and European genotypes [8]. The *Fhb5* gene locates close to the centromere, and it is proved to be linked with important agronomical traits; however, the resistance source 'Wangshuibai' carries unfavorable alleles in this respect. The *Qfhs.ifa-5A* locus originated from the 'CM-82036', conferring 20% of the phenotypic variance; however, the QTL effect was more pronounced after spray inoculation. Therefore, it is suspected that the QTL provided resistance more likely against the initial penetration. On the other hand, it was also proved that the QTL partially overlapped with a plant height affecting the region [57,58]. The location and effect of the identified QTL differed not only in the effect size but, more importantly, also in the location from the previously defined gene or QTLs. The AX-94387470 SNP marker sequence data analysis revealed that it bound to a region that coded a defense mechanism-related protein. Based on the PANTHER–Gene Information, the gene in Arabidopsis was determined as a coding cysteine-rich antifungal protein. Based on the allelic frequency, it could be concluded that the resistant parent 'BKT9086-95' carried a rare variation, which was present in the reference genotypes with 15.6% frequency. Based on the location, it is supposed that a novel resistance QTL was identified in the 'BKT9086-95' line, which could contribute to the further characterization and understanding of the genetic nature and resistance mechanisms against Fusarium head blight pathogens.

## 5. Conclusions

The Fusarium head blight type II resistance of lines originating from the 'BKT9086-95/Mv Magvas' crossing was investigated under greenhouse and field conditions. Based on the ANOVA test, it was determined that the genotype had a significant effect both on the infection values recorded 21 DAI and the size of the area under the disease progress curve.

According to our results, the other plant morphological and phenological characteristics had a smaller effect on the AUDPC values; therefore, during our mapping works, we found them more suitable for comparing the resistance levels of the lines. It was found that the sequence of the genotypes did not change in the individual repetitions with statistical proof. Therefore, it was confirmed that the lines might carry genetically encoded resistance.

Consistently expressed QTL was identified on the chromosomes 2A, 2B, 2D, 5A, and 7A in all years and in both testing systems (greenhouse and field) in the case of infection values 21 DAI.

In connection with the AUDPC values, the QTLs with the consistent expression were identical with the ones identified in the case of the infection percentages; however, except for chromosome 5A, they were restricted to a narrower region.

During the investigation of the other traits influencing the head blight resistance (plant height, heading time, ear compactness) and the molecular background of the resistance QTLs, the total overlap was found in the case of the region identified on chromosome 2D and partial overlap on chromosomes 2A and 2B.

In the case of the 'BKT9086-95' line, the presence of a DNA section encoding a protein linked to the plant protection mechanism was suspected on chromosome 5A. Based on the analysis of the allele type, it was found that a rare version was identified.

**Supplementary Materials:** The following are available online at http://www.mdpi.com/2073-4395/10/8/1128/s1, Figure S1: Changes in yearly precipitation (a) and average temperatures (b) in the experimental years obtained from long-term average data (1981-2010) in Martonvásár, Figure S2: Correlation plot between plant height, ear compactness, date of infection and resistance, Figure S3: Quantile-quantile plots of the K model, Table S1: *p*-values of the significant SNP markers based on 21 DAI, Table S2: *p*-values of the significant markers based on AUDPC values.

**Author Contributions:** Conceptualization, E.V.-L. and K.P.; methodology, E.V.-L.; validation, G.V.; writing—original draft preparation and project administration, B.V.; writing—review and editing, K.P. and Z.F.; supervision and funding acquisition, O.V. All authors have read and agreed to the published version of the manuscript.

**Funding:** This research was supported by the Hungarian Government and the European Union, with the co-funding of the European Regional Development Fund in the frame of Széchenyi 2020 Program GINOP-2.3.2-15-2016-00029. The publication is supported by the EFOP-3.6.3-VEKOP-16-2017-00008 project. The project is co-financed by the European Union and the European Social Fund.

**Acknowledgments:** Special thanks are due to the colleagues of IFA Tulln, especially to Hermann Bürstmayr for the *Fusarium* isolates and their contribution to the microsatellite and AFLP detection. Special thanks are due to András Cseh for his contribution in the BLAST analysis.

**Conflicts of Interest:** The authors declare no conflict of interest. The funders had no role in the design of the study; in the collection, analyses, or interpretation of data; in the writing of the manuscript, or in the decision to publish the results.

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
