# Peer review of "Analysis of Genetic Factors Defining Head Blight Resistance in an Old Hungarian Wheat Variety-Based Mapping Population"

_agronomy, doi:10.3390/agronomy10081128_

Round 1 Peer-Review

Reviewer 1 Report

Please see the file with comments attached.

Author Response

First and foremost, we would like to express our thankfulness for the thorough revision and the insightful comments

Responses to the reviewer's comments:

Any specific reason why Fusarium damaged kernels and DON content were not studied?

Mycotoxin studies and kernel infection resistance determination require a higher amount of infected materials, which usually is not the aim of floret inoculation method due to its labour-intensive nature. For this purpose, different inoculation method should be applied. Although we are aware that moderately resistant genotypes tend to accumulate a higher amount of mycotoxin, the available seed lot made the determination of mycotoxin and kernel resistance impossible. On the other hand, we also see the relevance of determining Fusarium-damaged kernels and DON content and agree that it would be a promising research direction in the future.

Please explain why only 175 (173 + 2) lines from a population of 250 were studied. What criteria was used to reduced the population size?

Lines which had more than 3 missing data points under greenhouse or field conditions over the experimental years were excluded from the analysis.

Type II resistance, while critical, is not the only factor which causes yield and quality losses. In a favourable year, numerous point infections can lead to major losses. Although the conclusions drawn based on the experiments look to be sound, there is a certainly a bigger discussion required so that the effect of FHB is not over-simplified.

In our research, we aimed to prove that the selected line derived from the old Hungarian wheat variety carries genetically coded resistance; however, type II is an important component of the resistance and could be tested accurately. We would like to extend the analysis and include other components as well in order to obtain a clearer picture of the nature of the resistance and involve more old varieties into the analysis. However, that would fall under the scope of a forthcoming experiment.

Introduction lacks information on FHB resistance. If the markers are already known for various sources of resistance and the authors are testing them again, what novel information do they hope to capture for FHB resistance, which is a definitely a quantitative trait.

’BKT9086-95’ line was derived from an old variety, it expressed a promising level of resistance under routinely conducted artificial inoculation test in the test nursery of the Centre for Agricultural Research. By the testing of the reported SSR markers, our aim was to identify whether the line carried QTL at an already identified location and also to determine the allele type. In addition, identifying a set of markers applicable for marker-assisted selection was also one of the goals.

In the year 2020, this is below the threshold number of markers to be used in wheat for generating a linkage map.

Thank you for the remark, we have no objection to it. The ANOVA test was performed for this the reason: to determine whether any of the tested markers could be associated with the trait of our interest, before performing SNP analysis.

The basis of marker elimination to find the reliable markers is not sound. Since the marker density was too low to start with, one would wonder how these markers can be reliably used in breeding programs based on their linkage to the trait.

At the beginning of the experiment, our original attempt was to identify markers for practical breeding. However, SNP markers became more and more affordable over time. Therefore, the potential association was used as the threshold for further study with more robust methods.

Was the SNP array data used for mapping?

Yes, it was.

The FHB resistance data from the population should be part of the main manuscript.

’BKT9086-95’ line  derived from ’Bánkúti 1201’ had been tested previously, and those results were already published in a referred journal (László, E.; Puskás, K.; Vida, G.; Bedő, Z.; Veisz, O. Study of Fusarium head blight resistance in old Hungarian wheat cultivars. Cereal Res. Commun. 2007, 35, 717–720). In that experiment, different old Hungarian wheat varieties were tested (e.g.: Székács, Diószegi) Those varieties were more advanced than landraces, yet they were more heterogeneous than modern wheat varieties. ’Bánkúti 1201’ was dissected to lines based on their storage protein composition (Juhász et al. Bánkúti 1201—an old Hungarian wheat variety with special storage protein composition. Theor. Appl. Genet. 107, 697–704 (2003). https://doi.org/10.1007/s00122-003-1292-2), and they became the subjects of FHB testing.

Is the summary of the manuscript "FHB resistance is genetic and most of the markers confirmed in this study are previously known"?

Old Hungarian varieties are tall (up to 145 cm); therefore, it was not clear whether the resistance was a result of resistance or escape mechanisms. In the case of the studied lines, the identified QTL on chromosome 5A is a novel one and the allele type is rare. It differs not only in the effect size, but more importantly, also in the location from the previously defined gene or QTLs. The AX-94387470 SNP marker sequence data analysis revealed that it is bound to a region which coded a defence mechanism-related protein. Based on PANTHER – Gene Information, the gene in Arabidopsis was determined as a coding cysteine-rich antifungal protein. Based on the allelic frequency, it could be concluded that the resistant parent ’BKT9086-95’ carried a rare variation, which was present in the reference genotypes with 15.6% frequency. Based on the location, it is supposed that a novel resistance QTL was identified in the ’BKT9086-95’ line, which could contribute to the further characterisation and understanding of the genetic nature and resistance mechanisms against Fusarium head blight pathogens.

Height and days to flowering are critical under natural field inoculation conditions for FHB damage. This conclusion can be mis-leading.

The mentioned statement was revised, and the authors agree with the comment. Therefore, the statement has been removed from the conclusions.

Reviewer 2 Report

I have several comments addressed to the authors regarding the clarity of methods and results.

In my opinion the section “Artificial inoculation” should be supplemented by answering the following questions:

At what exact growth stage (BBCH or other scale) wheat heads were inoculated? Were the florets injected with without or wounding (using syringe or pipet)?

How many plants where taken per treatment (replication) at the field and greenhouse experiments?

Were all the lines and parents flowered at the same time? Was the inoculation of all genotypes performed at the same day?

Lines 138-139. If authors would mention the actual long-term average values of temperature and precipitation, it would show a real picture of the weather conditions during the experimental time.  

Section “Molecular methods” Line 153.  Authors could explain when the plants were collected for the DNA extraction.

To my mind, in lines 203-206 and 229-244 the authors present a discussion rather than results, therefore I suggest moving this text to the discussion part.

Line 222, 224 and 226. What does the authors have in mind by terms "infection time", " different number of repetitions" and "date of inoculation"? There was nothing mentioned about such conditions in "Materials and methods", therefore it needs some explanation.

The literature source needs to be moved back in line 420, i.e. “Based on our previous results [36] and our three-year investigation…”

Author Response

First and foremost, we would like to express our thankfulness for thorough revision and insightful comments.

Responses to the reviewer's comments:

At what exact growth stage (BBCH or other scale) wheat heads were inoculated? Were the florets injected with without or wounding (using syringe or pipet)?

The growth stage was BBCH 61. Repeating pipette was used for injection and wounding was also applied.

How many plants where taken per treatment (replication) at the field and greenhouse experiments?

The inoculated heads were handled as repetition. In greenhouse, 4 heads were taken in each experimental year, whereas under field conditions, 10 heads were inoculated yearly.

Were all the lines and parents flowered at the same time? Was the inoculation of all genotypes performed at the same day?

The lines flowered in 2 weeks interval (average over the experimental years). Artificial inoculation was carried out not on the same day, but when the plants reached the same developmental stage (BBCH 61). Repeating pipette was used for injecting spikelet, with wounding.

Lines 138-139. If authors would mention the actual long-term average values of temperature and precipitation, it would show a real picture of the weather conditions during the experimental time. 

The WMO recommended to use standard climate periods as controls; these are 30-year long periods. For example, from 1971-2000, 1981-2010. Data for 2020 are not available, hence we used the actual WMO standard.

Section “Molecular methods” Line 153. Authors could explain when the plants were collected for the DNA extraction.

In the first greenhouse experiment, the flag leaves of the non-inoculated shots were taken for DNA analysis from the first repetition.

To my mind, in lines 203-206 and 229-244 the authors present a discussion rather than results, therefore I suggest moving this text to the discussion part.

Although the authors see the reviewer’s point, in this case, we would prefer to leave these statements in the Results chapter, as they explain the logical steps leading up to the genetic studies.

Line 222, 224 and 226. What does the authors have in mind by terms "infection time", " different number of repetitions" and "date of inoculation"? There was nothing mentioned about such conditions in "Materials and methods", therefore it needs some explanation.

Analysis of variance and correlation calculation was applied for testing the relationships between 21 DAI, AUDPC and plant height, inoculation time (beginning of flowering) and ear compactness. Under field conditions, in 2009, only the Fusarium graminearum inoculation was evaluated; therefore, the number of repetitions differed between years.

The literature source needs to be moved back in line 420, i.e. “Based on our previous results [36] and our three-year investigation…”

Thank you for the comment. Literature has been moved to the introduction of the plant materials.

Reviewer 3 Report

I have reviewed the manuscript entitled “Analysis of Genetic Factors Defining Fusarium Head Blight Resistance in an Old Hungarian Wheat Variety Based on Mapping Population” submitted to the journal of Agronomy. The authors evaluated a set of 250 F6 recombinant inbred lines (RILs) from a bi-parental cross BKT9086-95/Mv Magvas for their reactions to fusarium head blight (FHB). The bi-parental population was tested under greenhouse and field conditions for three years. The parents and the RILs were genotyped using SSR, AFLP, and SNPs. The polymorphic markers were used for mapping FHB resistance QTL.

Major points:

  • The authors used inappropriate analysis to identify the FHB QTL: This study used genome-wide association study (GWAS) to identify QTL in a bi-parental population of 250 RILs. GWAS is used on natural population or nested association mapping but not on a bi-parental population.
  • The authors mentioned in line 250-251 that the markers used did not provide good coverage of the whole wheat genome. However, good genome coverage should be important for QTL mapping.
  • In materials and methods: the authors did not provide enough data on the markers used in mapping (e.g. the coverage per chromosome), was the inoculation in the field similar to that in the greenhouse? Were all wheat heads in the field injected with the conidial suspension in the field? What is the type of markers agat17, gtac2 and gtac3
  • Many English issues in this manuscript. In addition there were some inappropriate terminology (e.g. wheat ear, partial/total overlapping, total square, “head blight” in the abstract, agronomic characters, homogeneous sister lines, homogeneity of deviation).
  • Table 1 and 3: It would be better to present the phenotypic data distributions as figures. Figure1 can be used as supplementary

Author Response

First and foremost, we would like to express our thankfulness for thorough revision and insightful comments.

Responses to the reviewer's comments:

The authors used inappropriate analysis to identify the FHB QTL: This study used genome-wide association study (GWAS) to identify QTL in a bi-parental population of 250 RILs. GWAS is used on natural population or nested association mapping but not on a bi-parental population.

The applied methodology was chosen according to the following literature: Rasheed et al, 2017 (Awais Rasheed, Yuanfeng Hao, Xianchun Xia, Awais Khan, Yunbi Xu, Rajeev K. Varshney and Zhonghu He.: Crop Breeding Chips and Genotyping Platforms: Progress, Challenges, and Perspectives, Molecular Plant, Volume 10, Issue 8, 1047 – 1064. (“GWAS in natural diversity panels, and joint linkage-association mapping using both bi-parental and natural populations”)

The authors mentioned in line 250-251 that the markers used did not provide good coverage of the whole wheat genome. However, good genome coverage should be important for QTL mapping.

The AFLP and SSR marker coverage do not allow to perform a QTL analysis. The genetic map was fragmented, but SNP method became more affordable over time for plant genomic studies. Therefore, on the SSR and AFLP marker sets, a simple marker-trait association was carried out and the results served as a benchmark to move forward to more robust technics such as SNP (by Illumina Infinium 20k wheat chip).

In materials and methods: the authors did not provide enough data on the markers used in mapping (e.g. the coverage per chromosome), was the inoculation in the field similar to that in the greenhouse? Were all wheat heads in the field injected with the conidial suspension in the field? What is the type of markers agat17, gtac2 and gtac3

As a first step, the differences between the parents were analysed with the selected SSR and AFLP primers [29-30, Somers et al., 2004]. The selected SSR markers covered the wheat genome in 20-40 cM intervals. The same method was used for artificial inoculation under greenhouse and field conditions. In the greenhouse, 4 heads of each line were injected with Fusarium culmorum, and in the field, 5-5 heads were inoculated by Fusarium graminearum and Fusarium culmorum isolates, respectively. agat17, gtac2 and gtac3 markers are derived from the AFLP analysis.

Many English issues in this manuscript. In addition there were some inappropriate terminology (e.g. wheat ear, partial/total overlapping, total square, “head blight” in the abstract, agronomic characters, homogeneous sister lines, homogeneity of deviation).

The manuscript has been revised by a professional English proofreader.

Table 1 and 3: It would be better to present the phenotypic data distributions as figures. Figure1 can be used as supplementary

Phenotypic distribution has been added as a supplement, just like weather data of the experimental years.